# Responding to the impact of COVID-19 on the mental health and well-being of health workers in LMICs

Jerilyn Hoover[1] , Paul Bolton[2], Ashley Clonchmore[3] , Linda Sussman[4] and Diana Frymus[1]

[1]Office of HIV/AIDS, U.S. Agency for International Development, Washington, DC, USA; [2]Inclusive Development Hub, Bureau for Development, Democracy, and Innovation, U.S. Agency for International Development, Washington, DC, USA; [3]Office of HIV/AIDS, Credence Management Solutions, LLC, Global Health Training, Advisory, and Support Contract at the U.S. Agency for International Development, Vienna, VA, USA and [4](Formerly) Office of Population and Reproductive Health (PRH), Bureau of Global Health, U.S. Agency for International Development, Washington, DC USA

## Perspective

**Keywords:**
healthcare workers; health care system; global health; mental health; occupational health

**Corresponding author:**
Ashley Clonchmore;
Email: aclonchmore@usaid.gov

### Abstract

The COVID-19 pandemic has worsened mental health among health workers around the world. With a projected global shortage of 10.2 million health workers by 2030, further exacerbated by COVID-19, taking action to support health worker mental health needs to be an integral component of investments to overcome this gap and build resiliency of systems for the future. Health workers are functioning in highly stressful environments at great personal risk to provide services that improve quality of life and save lives. To reduce burnout and early exits from the workforce, health workers must be protected and equipped to work in supportive environments, manage stress, and access mental health services when needed. This article explores the impact of COVID-19 on health worker mental health and proposes actions for health systems and workplaces to support health workers which draw on available evidence and examples of USAID-supported partner activities.

### Impact statement

The COVID-19 pandemic has worsened the mental health of health workers globally. As we reflect on lessons learned from the pandemic, this is an important time for dedicated action on mental health to strengthen and protect the health workforce. This article recommends actions for employers of health workers and country health systems based on current evidence and informed by examples from global programs implemented by USAID-supported partners during the pandemic. These actions are to:

– Improve the environment to increase workplace safety and reduce psychosocial risks.
– Provide information to support the development of stress coping skills.
– Identify persons with mental health conditions and support access to mental health treatment.
– Promote inclusion of mental health treatment into occupational health policies.

### Introduction

In 2020, at the beginning of the COVID-19 pandemic, there was a global outpouring of support and appreciation for health workers and other essential workers, with nightly claps, cheers, and noise making among the displays of gratitude (Snouwaert, 2020). The World Health Organization (WHO), in recognition of the dedication of health workers around the world, designated 2021 as The Year of the Health and Care Worker (*2021 Designated as the International Year of Health and Care Workers*, 2020). Along with a resolution at the 2022 World Health Assembly, the WHO launched a health and care worker compact, which "provides technical guidance for Member States and relevant stakeholders on how to protect and safeguard the health, safety, and human rights of health and care workers everywhere, and ensure they have safe, supportive enabling work environments" (*Human Resources for Health – WHA 75.17*, 2022). The International Labor Organization (ILO), at its annual conference in 2022, added occupational safety and health as the fifth *Fundamental Principles and Rights at Work,* affirming a duty of care to support health worker health, mental health, and well-being (*International Labour Conference Adds Safety and Health to Fundamental Principles and Rights at Work*, 2022). The ILO and WHO have also newly released guidelines and a policy brief on *Mental Health at Work* (*WHO Guidelines on Mental Health at Work*, 2022). Other recent commitments, including several from the U.S. government, align and intersect with these efforts. The U.S. Government Global Health Worker Initiative was launched by the

White House in May 2022 in recognition of the global contributions of health workers during the pandemic and the barriers they face. One of the four pillar areas of the initiative is protecting health workers, which includes expansion of support and services for mental health as well as physical health and well-being (*FACT SHEET: The Biden-Harris Administration Global Health Worker Initiative*, 2022). This priority is further integrated into the U.S. Department of State's COVID-19 Pandemic Prioritized Global Action Plan with a core line of effort to support the health, safety, well-being, and effectiveness of frontline health workers (*COVID-19 Pandemic Prioritized Global Action Plan for Enhanced Engagement (GAP)*, n.d.). USAID also has a forthcoming agency mental health policy that will include discussion of worker mental health, given the impact it has on quality of care and capacity for mental health service delivery. The growing recognition and prioritization of support for the mental health and well-being of our health workers presents an opportune time to leverage recognition into action. Particularly in low- and middle-income countries, where there is chronically low availability of formal mental health services and where under resourced health systems place additional burden on the health workforce, governments and organizations employing health workers need to invest in strengthening systems that support prevention of burnout[1] and address ongoing threats to mental health (*Mental Health Atlas 2020*, 2021). As the authors are global health practitioners focused on health workforce, mental health, and psychosocial support issues, this piece explores the impact of COVID-19 on health worker mental health. It proposes actions for the global community to enact system-wide reforms across health systems and within workplaces that support individual health workers based on available evidence and informed by example activities from USAID-supported partners.

## Impact of COVID-19 on health workers

To build and maintain strong health systems that deliver quality healthcare, it is essential to support the mental health needs and wellness of health workers. The COVID-19 pandemic has caused major disruptions across countries' healthcare systems. Health workers have experienced increased workloads, risks to personal safety, and ongoing operational challenges. Weak support systems to navigate difficult and rapidly changing work environments have created conditions that negatively impact the mental health of health workers, which was already inadequately addressed prior to the pandemic. Globally, there is a lack of worker mental health and psychosocial support services (*Mental Health Atlas 2020*, 2021, p. 136). Risks to personal safety have been a major contributor to workplace related stress and anxiety. Workplace safety stressors have included an increase in violence toward health workers (with 3,400 recorded attacks on health workers or health facilities since December 2019), fear of contracting COVID-19 with long-term health impacts, and anxiety about exposing families and loved ones (Devi, 2020; *Attacked and Threatened: Health Care at Risk*, 2022). Health workers have experienced an increase in stress and burnout related to caring for extremely sick and dying patients, often with insufficient space, personal protective equipment (PPE), and staffing (Gotinga, 2021; Kaur, 2021; Noguchi, 2021). In settings, where mental health and psychosocial support services are available, high-pressure working environments often limit ability to engage those services. Health workers may also fear stigma and discrimination from colleagues or supervisors and

risk to their licensure for seeking mental health care (Knaak et al., 2017; Gold, 2020). These pandemic impacts and mental health effects have been disproportionately borne by women, who make up 70% of the global health and care workforce, and by staff who are earlier in their careers or are in lower-status positions and have less power to negotiate their working conditions (Serrano-Ripoll et al., 2020).

## Effects of poor health worker mental health

Poor mental health has both personal and professional effects, which affect health systems and the quality of care. During the pandemic, health workers have experienced higher rates of depression, anxiety, insomnia, and distress that have contributed to high rates of burnout and early exits from the workforce (Muller et al., 2020; Ibrahim et al., 2022). Surveys among U.S. healthcare workers found that nearly one in five had quit their jobs during COVID-19 (Galvin, 2021). Additionally, 66% of nurses working in acute and critical care said their experiences during the pandemic have made them consider leaving their profession (Galvin, 2021; Viejo, 2021). The International Council of Nurses predicts that globally, at least 10% of nurses will leave the profession due to burnout and exhaustion (*WHO Media Briefing on COVID-19*, 2021). Given the projected global shortage of 10.2 million health workers by 2030, we should do everything possible to support and retain our current health workers (*WHO Health Workforce*, n.d.). In recent years, researchers and practitioners have increasingly recognized the *moral injury* often underlying health worker burnout, which can be defined as "knowing what a patient needs but being unable to provide it because of constraints beyond one's control" (Talbot and Dean, 2018; Dean, 2021). Health workers experience moral injury when they are required to work in extremely challenging conditions for long periods of time with insufficient staff, supplies, or protective measures. This is a violation of workers' rights and is both unethical and unsustainable (*COVID-19: Occupational Health and Safety for Health Workers: Interim Guidance*, 2021). Health workers may also experience vicarious trauma and develop compassion fatigue, which are associated with exhaustion and burnout (Lluch et al., 2022). As there is a relationship between health worker burnout or poor health and medical errors, high-quality health care depends, in part, on the well-being of the health workers delivering it (Mazurek Melnyk et al., 2021). A study among health providers working in maternity units in Malawi found that burnout and depression were related to lower levels of respectful maternity care (Zieman, 2021). They also found that supportive relationships between health workers and their facility managers can protect against burnout and improve care. Other literature supports these findings, noting that the well-being of health workers impacts how they treat clients, which in turn can affect clients' care seeking behaviors and overall experience of care (Kruk et al., 2009; Bohren et al., 2017; Manning and Schaaf, 2018). Ensuring health workers receive adequate support and services for mental health needs to be an essential part of protecting the health workforce globally. This is also critical for the future of the health workforce, as those experiencing the heaviest burden of mental health effects (women and individuals early in their careers) represent the majority of the global health workforce and are the next generation of health workers and leaders.

## Suggested interventions to protect and support health worker mental health

### Improve the environment to increase workplace safety and reduce psychosocial risks

COVID-19 has increased stress in both the work and home environments of health workers (*Health Workforce Policy and*

---

[1] We use "burnout" consistent with the ICD11 classification of it as an occupational phenomenon, and not as a medical condition. See https://icd.who.ent/en.

*Management in the Context of the COVID-19 Pandemic Response: Interim Guidance*, 2020). The pandemic has increased workloads and extended working hours, causing stress and fatigue (*COVID-19: Occupational Health and Safety for Health Workers: Interim Guidance*, 2021). While patient numbers are outside of the full control of the health system, strict attention to balancing workloads among staff is important, not only to reduce stress but also so that all workers feel they are fairly treated (*Health Workforce Policy and Management in the Context of the COVID-19 Pandemic Response: Interim Guidance*, 2020). This can be done through regular discussions between managers and staff on workflow and time management to ensure realistic workload expectations. Long shifts increase exhaustion and stress which contribute to clinical errors and risk to staff and patients (Rosenberg, 2014; *Long Work Hours, Extended or Irregular Shifts, and Worker Fatigue*, n.d.). While extended hours and extra shifts may be unavoidable during short periods of increased demand, this type of scheduling is not sustainable long term. To help combat stress from high workloads, it is important to provide health workers with sufficient time between shifts to recharge by leaving the premises and completely separating from the work environment (*Health Workforce Policy and Management in the Context of the COVID-19 Pandemic Response: Interim Guidance*, 2020).

Other strategies to reduce stress include introducing flexible scheduling to shift workers with self-identified health risks from patient-facing to non-patient facing roles, if needed and agreed upon, and identifying innovative ways to deliver services to reduce risk to health workers (Frymus and Daniels, 2021). Any decisions related to the work environment should be made in consultation with the health workers involved. Health workers should be clearly made aware of their rights and supported to set and maintain boundaries to protect their well-being. Health workers' contributions should be appreciated and appropriately compensated through on-time payment, including any additional risk allowances or incentives for the COVID-19 pandemic. Governments and organizations employing health workers can reduce workers' health risks by ensuring sufficient PPE is available and prioritizing health workers in ongoing vaccination efforts (*Health Workforce Policy and Management in the Context of the COVID-19 Pandemic Response: Interim Guidance*, 2020). This can support them to engage with family and friends outside work, which is an important part of stress management. See Table 1 for examples of changes USAID partners made to improve work environments for health workers (Frymus and Daniels, 2021; *Partner Operational Solutions in Response to COVID-19: Caring for Healthcare Workers and IP Staff During the COVID-19 Pandemic*, 2021). Some partners provided feedback that interventions recognizing and supporting health workers as individuals increased morale, commitment to work, and sense of belonging.

### Provide information to support development of stress coping skills

While changes to the work environment are a direct cause of stress, stress itself is the result of how we respond to a stressful situation. Equipping health workers with knowledge of how stress impacts them is helpful in reducing stress. This can be provided through training, which emphasizes that changing our mental state in positive ways is possible – this is referred to as a growth mindset approach. When combined with basic skills to reduce stress, this can have a substantial positive effect. Examples of these skills include cognitive coping, which is a process to identify unhelpful thoughts and their links with stress and distressing

**Table 1.** Examples of changes to improve work environments, implemented by USAID-supported partners

| Example | Country(ies) of partner example |
|---|---|
| Providing PPE for staff and following infection prevention and control practices | Nigeria, South Africa |
| Creating flexible and staggered scheduling to reduce COVID-19 risk and stress levels among health workers | Eswatini, Nigeria |
| Relocating some services to be provided outside of facilities | South Africa |
| Accommodating health workers with higher risk of health complications from COVID-19 to deliver virtual rather than in person services, reducing their risk | South Africa |
| Increasing health insurance and life insurance benefits for staff to improve financial security | Morocco, Nepal, Nigeria, South Africa |
| Publicly recognizing and affirming the contributions of staff through patient appreciation boards and awards | South Africa, Nepal, Nigeria |

**Table 2.** Actions by USAID-supported partners to support health worker stress management during COVID-19

| Example | Country(ies) of partner example |
|---|---|
| Hold group support sessions | South Africa |
| Survey staff stress levels and coping mechanisms | Lebanon, South Africa |
| Create team building environments to support team bonding | Nigeria |
| Offer counselors for stress management | Tunisia, South Africa |
| Provide psychosocial self-care sessions and support for staff | Eswatini, Malawi, Nepal, South Africa |
| Provide resiliency training for staff, including communication strategies in difficult and stressful situations* | India |
| Provide psychological first aid and other mental health training on identifying signs of burnout and managing psychological reactions to stress | Eswatini, Lebanon, South Africa |

*As discussed previously, any interventions for individual health workers should be paired with activities to reduce stress and improve conditions in the overall work environment.

emotions and replace them with less stressful ways of thinking. Relaxation techniques and mindfulness can help maximize recovery time in breaks during work and between shifts. These concepts have been incorporated into references such as WHO's "*Doing What Matters in Times of Stress: An Illustrated Guide*" as well as short videos, apps, and media for both health workers and the general population (*Doing What Matters in Times of Stress: An Illustrated Guide*, 2020). In most countries, health workers have access to one or more of these information sources. Table 2 outlines a number of additional interventions to improve health worker's stress management implemented by USAID-supported partners (*Partner Operational Solutions in Response to COVID-19: Caring for Healthcare Workers and IP Staff During the COVID-19 Pandemic*, 2021).

### Identify persons with mental health conditions and support access to mental health treatment

Collectively, we consider approaches to reduce stress as "Psychosocial Support" or PSS (YouthPower, n.d.). These include the most commonly and broadly implemented mental health and psychosocial support activities because they are easily disseminated. While they are important in reducing stress and can help prevent stress-related mental health conditions, they are typically not treatments for more severe mental health conditions once they occur. This includes depression, anxiety, posttraumatic stress disorder, and substance abuse. These conditions require psychotherapy and/or medications and can have serious consequences if untreated, such as suicide, domestic violence, and child abuse (*Depression*, 2021; *WHO Releases Guidance on Mental Health Care after Trauma*, 2013). For this reason, PSS activities should include information on how to identify mental health conditions in self and others and how and where to seek treatment. Employers of health workers should ensure that managers are trained to recognize warning signs and provide or refer staff to confidential treatment services through employee assistance programs or similar offerings, providing time away from work to seek services. Where worker numbers are sufficient and privacy allows, group treatment can be highly effective and build a supportive group of peers (*Group Interpersonal Therapy (IPT)*, 2020; *Community-Based Mental Health Services Using a Rights-Based Approach*, 2021). Because of the stigma associated with mental health conditions and the greater resistance to treatment among health workers, information on these disorders should emphasize their impact on personal well-being and family, as well as professional impacts, and highlight the effectiveness of treatment. Additional examples of actions taken by USAID partners to support health worker mental health and treatment access are listed in Table 3 (*Partner Operational Solutions in Response to COVID-19: Caring for Healthcare Workers and IP Staff During the COVID-19 Pandemic*, 2021). Partners who provided these examples reflected that health worker and client mental health is inextricably linked, that COVID-19 helped catalyze action to address long-standing mental health needs, and that they saw some improvements in clinical outcomes when health worker and client mental health was supported.

**Table 3.** Examples from USAID-supported partners of activities to improve health worker mental health and access to treatment services

| Example | Country(ies) of partner example |
| --- | --- |
| Routine supervisor check ins on staff | Nigeria |
| Provide counseling for staff and acknowledgement of the stress of supporting clients in highly difficult situations | Egypt, Nepal, South Africa, Tunisia |
| Provide screening and treatment (or referrals) for mental health concerns | El Salvador, Eswatini, Guatemala, Honduras, Kenya, Malawi, South Africa |
| Facility activities to support health worker mental health, including group debriefing sessions | South Africa |
| Use telehealth to expand access to mental health services for staff | South Africa |

### Promote inclusion of mental health treatment into occupational health policies

In addition to interventions to make work environments safer and less stressful, support health workers to cope with stress, and provide access to mental health treatment when needed, it is critical to ensure that countries' occupational health policies include mental health treatment services as essential for protecting and supporting health workers. It is the mandated obligation of countries to ensure occupational health policies require safe working environments free from harassment and violence, regular scheduled breaks and limits on shift length, maximum patient to staff ratios, fair and timely compensation, and comprehensive health benefits, including mental health treatment, support to return to work, and reasonable accommodations for health workers with mental health conditions. Health workers and their representatives should be engaged when developing policies or guidance which would impact them and their work. There should also be collaboration with health professional licensing boards to ensure that accessing mental health services does not negatively impact licensure or ability to practice. These policies are necessary for incorporating interventions across a health system as formal benefits and resources for health workers and can encourage employers to provide dedicated time for their use.

### Conclusion

Prioritizing the mental health of health workers around the world is needed to have well-functioning healthcare systems. Recent global efforts to recognize the health workforce and commit to protecting and equipping them should be followed by dedicated investments and actions to support health worker mental health at individual, workplace, and systems levels. These efforts should be well coordinated and documented to maximize their impact, as more detailed documentation of mental health interventions for health workers in low- and middle-income country settings will increase understanding of barriers and build the evidence base for mental health and protective interventions in a variety of contexts. The negative impact of the COVID-19 pandemic on mental health and a growing global health worker shortage have highlighted a potential healthcare crisis and the time to act is now.

**Open peer review.** To view the open peer review materials for this article, please visit http://doi.org/10.1017/gmh.2023.30.

**Acknowledgments.** We would like to thank Mark Van Ommeren, Aiysha Malik, and Meredith Fendt-Newlin from the WHO who reviewed and provided comments on the manuscript. We would also like to thank Emily Reitenauer and Caroline Kasman who provided additional comments on the draft.

**Author contribution.** All authors contributed to the idea for the article. JH, PB, AC, and LS drafted the article. All authors reviewed and edited the document and approved the final manuscript.

**Financial support.** There was no funding source for the article.

**Competing interest.** The authors declare no competing interest.

**Disclaimer.** The views in this article are those of the authors and do not necessarily reflect the view of the U.S. President's Emergency Plan for AIDS Relief, the U.S. Agency for International Development, or the U.S. Government.

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
