## [Reviewer Report]

February 9, 2023

Drs. Gary Belkin, Judy Bass, and Dixon Chibanda

Editorial board leadership 

Cambridge Prisms: Global Mental Health 

To Drs. Belkin, Bass, and Chibanda, 

I am pleased to submit our paper titled “Responding to the impact of COVID-19 on the mental health and well-being of health workers in LMICs” for consideration as a perspective piece in Cambridge Prisms: Global Mental Health. We highlight the toll that COVID-19 has taken on the mental health and well-being of health workers globally, and the impact this has on individual health workers and on health systems overall. We propose key actions to capitalize on the growing momentum around worker mental health to support the mental health and wellness of health workers, particularly critical in the context of a 10.2 million projected health worker shortage by 2030.

This article is timely given several high profile developments last year, including the WHO Health and Care Worker Compact, launch of the Biden Administration’s Global Health Worker Initiative, the recognition by the International Labor Organization of occupational safety and health as a fundamental right, and the joint WHO and ILO guidelines on Mental Health at Work. Through submitting our article to Global Mental Health, we hope to raise the importance of leveraging recognition into action to support the health workforce that has worked valiantly to ensure delivery of essential health services during the pandemic.

Sincerely,

Jerilyn Hoover, BSN, MPH, RN

Health Workforce Development Advisor

Office of HIV/AIDS, Systems and Program Sustainability Division

U.S. Agency for International Development Washington, DC USA

Email: jehoover@usaid.gov Cell: +1 507-227-6233

---

## [Reviewer Report]

This in my opinion is a well written perspective paper - it is informative and provides practical recommendations to safeguard the mental health and well-being of HCWs in LMICs. I have not much to add, except perhaps two extra recommendations could be added in the manuscript: 

1. Actions to support health worker stress management during COVID-19 - time management training

2. Supporting health worker mental health and access to treatment services - Employee Assistance Program (EAP)

---

## [Reviewer Report]

• Overall, the paper could be further strengthened by focusing on USAID and PEPFAR-specific work.

• There already exists a substantial body of evidence and reviews of the evidence on the mental health impact on healthcare workers during COVID-19. Therefore, the first part of paper does not necessarily add much more than what we already know. If the goal of the paper is to respond to the impact of COVID-19 on HCWs, then it would be more useful to retain this as the main focus of the paper - especially in the context of the forthcoming USAID policy on mental health mentioned in Line 73. To this end, it would be very interesting to add more content around what the policy proposes to include and why (i.e., based on evidence on mental health burden and effect of existing interventions).

• Line 82-85: This sentence states that this paper, “...proposes actions for the global community to enact system-wide reforms across health systems and within workplaces that support individual health workers which draw upon USAID and PEPFAR-funded program experience over the past two years.” The rest of the paper needs to clearly conform to the aims of the paper as mentioned in this sentence.

• The section on Interventions could be re-worked to clearly describe USAID and PEPFAR-funded program experiences and then summarize lessons learned from these experiences into more insightful boxes/diagrams/flowcharts.

---

## [Reviewer Report]

Thanks to the authors for considering the suggestions provided for the earlier draft. This manuscript improves upon the earlier draft by providing examples of USAID and partner activities.